# Microalgal-Bacterial Granular Sludge Process in Non-Aerated Municipal Wastewater Treatment under Natural Day-Night Conditions: Performance and Microbial Community

**Shulian Wang [1], Lin Zhu [2], Bin Ji [2,*], Huan Hou [1] and Yingqun Ma [3,*]**

[1] School of Civil Engineering, Architecture and Environment, Hubei University of Technology, Wuhan 430068, China; wangshulian@hbut.edu.cn (S.W.); houhuankai@163.com (H.H.)

[2] Department of Water and Wastewater Engineering, Wuhan University of Science and Technology, Wuhan 430065, China; zhuwuster@163.com

[3] School of Chemical Engineering and Technology, Xi'an Jiaotong University, Xi'an 710049, China

* Correspondence: binji@wust.edu.cn (B.J.); Yingqun_Ma@xjtu.edu.cn (Y.M.)

**Abstract:** The microalgal-bacterial granular sludge (MBGS) process is expected to meet the future requirements of municipal wastewater treatment technology for decontamination, energy consumption, carbon emission and resource recovery. However, little research on the performance of the MBGS process in outdoor treatment was reported. This study investigated the performance of the MBGS system in treating municipal wastewater under natural alternate day and night conditions in late autumn. The results showed that the average removal efficiencies of Chemical oxygen demand (COD), $NH_4^+$-N and $PO_4^{3-}$-P on daytime before cooling (stage I, day 1−4) could reach 59.9% ± 6.8%, 78.1% ± 7.9% and 61.5% ± 4.5%, respectively, while the corresponding average removal efficiencies at night were 47.6% ± 8.0%, 56.5% ± 17.9% and 74.2% ± 7.6%, respectively. Due to the dramatic changes in environmental temperature and light intensity, the microbial biomass and system stability was affected with fluctuation in COD and $PO_4^{3-}$-P removal. In addition, the relative abundance of filamentous microorganisms (i.e., *Clostridia* and *Anaerolineae*) decreased, while *Chlorella* maintained a dominant position in the eukaryotic community (i.e., relative abundance > 99%). This study can provide a theoretical basis and technical support for the further engineering application of the MBGS process.

**Keywords:** microalgal-bacterial granular sludge; outdoor; wastewater treatment; microbial community succession

## 1. Introduction

Traditional activated sludge (CAS) process had been applied to treat municipal wastewater for more than ten decades. However, the CAS process typically consumes a large amount of energy for electrical aeration to oxidize organic matter and ammonia nitrogen, while it releases greenhouse gases (GHGs) and produces excessive sludge [1–5]. Thus, it is clear that the CAS process can hardly meet the increasingly demanding requirements in terms of energy consumption, carbon emissions and discharge water quality [6]. The microalgal-bacterial granular sludge (MBGS) process has been explored in various wastewater treatment processes. MBGS is a symbiotic microbial aggregate of microalgae (including simple cellular prokaryotes, eukaryotes and diatoms, etc.) and bacteria (including heterotrophic bacteria, phosphate-accumulating bacteria, etc.). In the MBGS process, microalgae can provide $O_2$ for the oxidation of organic matter, as well as consume $CO_2$ as a carbon source produced from bacterial respiratory [6,7]. Compared with CAS, the MBGS process owns the superb merits of low energy consumption, robust adaptability and resource recovery potential, which has been extensively reported [8–11].

Recently, it was found that the self-coupled MBGS process could remove respective 92.7%, 96.8%, 84.1% and 87.2% of organic matter, ammonia nitrogen, total nitrogen and

phosphorus from simulated municipal wastewater under a continuous light condition in 6 h with respective effluent concentrations of 18.3, 0.8, 3.9 and 0.4 mg/L, indicating the excellent potential of the MBGS process [12]. In addition, the MBGS process was highly adaptable to antibiotics and heavy metals in actual wastewater [13,14] thus was extensively used for the treatment of actual municipal wastewater [15,16]. However, the study of municipal wastewater treatment by MBGS under outdoor day-night alternating conditions was rarely reported, which restricted the further engineering application of this process.

This study aims to treat simulated municipal wastewater by MBGS process under outdoor day and night alternating conditions. The nutrient removal in daytime and night during weather changes was investigated. Moreover, the microbial response mechanisms of the MBGS process under varying temperatures and light were further discussed. It is expected this study can provide new insights into the performance of the MBGS process under alternate day and night conditions and offer a theoretical basis for its further application.

## 2. Materials and Methods

### 2.1. MBGS and Simulated Wastewater

MBGS used in this study was pre-cultured using seeding sludge derived from a wastewater treatment plant in Wuhan City, China. Briefly, seeding sludge was cultured in an SBR reactor, followed by inoculation using simulated municipal wastewater [17]. After granular sludge formation, they were cultured in a 500 mL conical bottle with continuous aeration under $200 \pm 10$ $\mu mol \cdot m^{-2} \cdot s^{-1}$ of LED light irradiation for one month. The mature MBGS with a size of $710.2 \pm 18.6$ $\mu m$ and a 5-min sludge volume index ($SVI_5$) of $73.6 \pm 10.1$ mL/g were obtained. The volatile suspended solids (VSS) concentration was about 3 g/L.

The main components of simulated municipal wastewater were 527.0 mg/L NaAc, 113.1 mg/L $NH_4Cl$, 30.3 mg/L $K_2HPO_4$, 50.0 mg/L $MgSO_4 \cdot 7H_2O$, 20.0 mg/L $CaCl_2$, 40.0 mg/L $FeSO_4 \cdot 7H_2O$ and 40.0 mg/L $NaHCO_3$, which were equivalent to 400.0, 30.0 and 5.0 mg/L of initial Chemical oxygen demand (COD), $NH_4^+$-N and $PO_4^{3-}$-P.

### 2.2. Experimental Setup

As shown in Figure 1a, 50 mL of glass anaerobic bottles were used to inoculate mature MBGS. The reaction devices were placed on the terrace of the Building of Engineering Training Center at Wuhan University of Science and Technology (longitude 116.31, latitude 39.98) from 10 September 2020 to 13 October 2020. The hydraulic residence time (HRT) was 12 h.

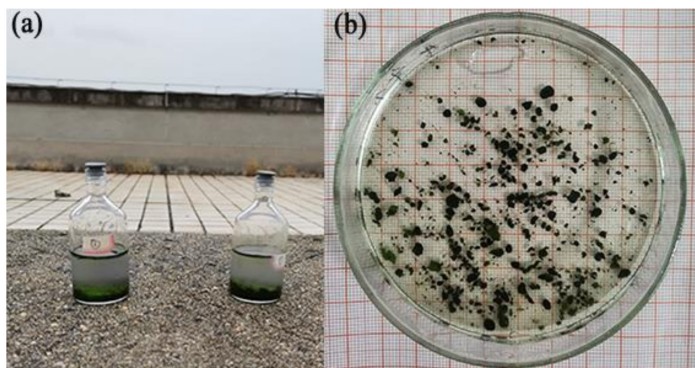

**Figure 1.** (**a**) A diagram of the experimental set-up and (**b**) the appearance of microalgal-bacterial granular sludge (MBGS).

### 2.3. Analytical Methods

During the experimental processes, 10 mL of the samples was taken from the bottles at daily 8 a.m. and at 8 p.m. Chemical oxygen demand (COD), ammonia nitrogen ($NH_4^+$-N), phosphate ($PO_4^{3-}$-P), nitrate nitrogen ($NO_3^-$-N), nitrite nitrogen ($NO_2^-$-N), VSS and $SVI_5$

were determined according to standard methods [18]. The concentration of chlorophyll a and b in MBGS were determined by the acetone extraction method [19]. The granular size of MBGS was measured by a particle size analyzer (Malvern Mastersizer 2000, Westborough, MA, USA). Dissolved oxygen (DO) concentrations and pH were measured using a dissolved oxygen meter (Yellow Springs, OH, USA) and pH meter (STARTER3100, Ohaus, Parsippany, NJ, USA). The daily ambient temperature, water temperature and light intensity were recorded. SPSS software was used to analyze statistically significant differences of MBGS performance ($p < 0.05$), and the Spearman correlation coefficient (ANOVA, $p < 0.05$) was used to assess the correlation between temperature, light intensity and nutrient removals.

The Illumina Miseq sequencing method was used to analyze the differences in the microbial community of sludge samples [20] on the 1st, 17th and 33rd days of the experiment. DNA was extracted using DNA kits (OMEGA Biotek Inc., Norcross, GA, USA) and verified with 0.8% agarose gel electrophoresis. The bacterial primers 338F-806R and algal primers 528F-706R were used to amplify gene fragments [21]. The sequencing process, including the data analysis, was performed by the approach described elsewhere [22].

## 3. Results

### 3.1. Performance of MBGS Process

The average ambient temperature and water temperature during the experiment were shown in Figure 2a. As seen, the average daily ambient temperatures were $24.0 \pm 1.5$ °C (phase I, day 1–4), $19.7 \pm 1.9$ °C (phase II, day 5–14), $21.6 \pm 0.9$ °C (Phase III, day 15–23), $13.8 \pm 1.6$ °C (phase IV, day 24–29) and $19.3 \pm 0.8$ °C (phase V, day 30–33). The average water temperature during the daytime was 0–3 °C higher than the average ambient temperature except for sunny days (i.e., day 2, 3, 9, 15, 18, 21, 22, 29–32), while the average water temperature at night was 0–3 °C lower than the average ambient temperature (Figure 2a). It should be noted that experimental water temperatures at daytime and night were highly variable due to changing ambient temperature. On the other hand, the light intensities during the experiment were presented in Figure 2b. It was found that the maximum light intensity was 680 $\mu mol \cdot m^{-2} \cdot s^{-1}$ at day 2–3, while 955 $\mu mol \cdot m^{-2} \cdot s^{-1}$ at day 9, 15, 18 and 22. On day 21 and 29–32, the maximum light intensity reached 1200 $\mu mol \cdot m^{-2} \cdot s^{-1}$. The rest experimental days were cloudy or rainy with light intensity ranging from 20 to 300 $\mu mol \cdot m^{-2} \cdot s^{-1}$.

The removal efficiency of the MBGS process in terms of COD, $NH_4^+$-N and $PO_4^{3-}$-P in municipal wastewater are presented in Table 1 and Figure 3. As can be seen, COD removal during the daytime was $44.1\% \pm 21.9\%$, while $31.9\% \pm 19.5\%$ at night, suggesting that bacteria in MBGS could accelerate the degradation of organic matters using $O_2$ produced from microalgae photosynthesis at daytime. COD removal in stage I was relatively stable due to the changeableness ambient temperature. While stage II experienced a drop in temperature, leading to decreased COD removal. In stage II at daytime, COD removal rate recovered from $32.9\% \pm 11.7\%$ to $63.2\% \pm 0.85\%$ (Figure 3a). Despite the increase of ambient temperature in Phase III, only sunny days (i.e., 4 days) provided sufficient light (Figure 2b), and thus, COD removal fluctuated sharply between $-29\%$ and 69%. In addition, the COD removal rate at night shifted at $-43\%$–55% due to the changes in ambient temperature. A negative COD removal rate was possible due to organic matter releases from microalgal or bacterial cells impaired the stable operation of the MBGS process. It should be noted that COD removal was temperature-dependent. Our previous study indicated MBGS process could perform satisfactorily in the temperature ranging from 15 to 30 °C, and the maximum COD removal was achieved at 30 °C [10]. However, it was found that COD removal was insignificantly correlated with average water temperature and light intensity (ANOVA, $p > 0.05$) in this study.

On the other hand, the average removal rate of $NH_4^+$-N at daytime was $77.5\% \pm 13.1\%$, while $46.9\% \pm 16.4\%$ at night with a significant difference ($p < 0.05$) (Figure 3b). $NH_4^+$-N removal at daytime was significantly related to the average water temperature (ANOVA, $p < 0.05$), but insignificant correlated with light intensity (ANOVA, $p > 0.05$), indicating

that water temperature may have a greater impact on $NH_4^+$-N removal than the light intensity at daytime. However, it should be found that $NH_4^+$-N removal reached the highest on sunny days (i.e., day 2, 3, 9, 15, 18, 21, 22 and 29–32) with light intensity above 680 $\mu mol \cdot m^{-2} \cdot s^{-1}$. These results indicated that high light intensity was more favorable for $NH_4^+$-N removal, but a low light intensity had an insignificant effect on $NH_4^+$-N removal, which was consistent with reported research [23].

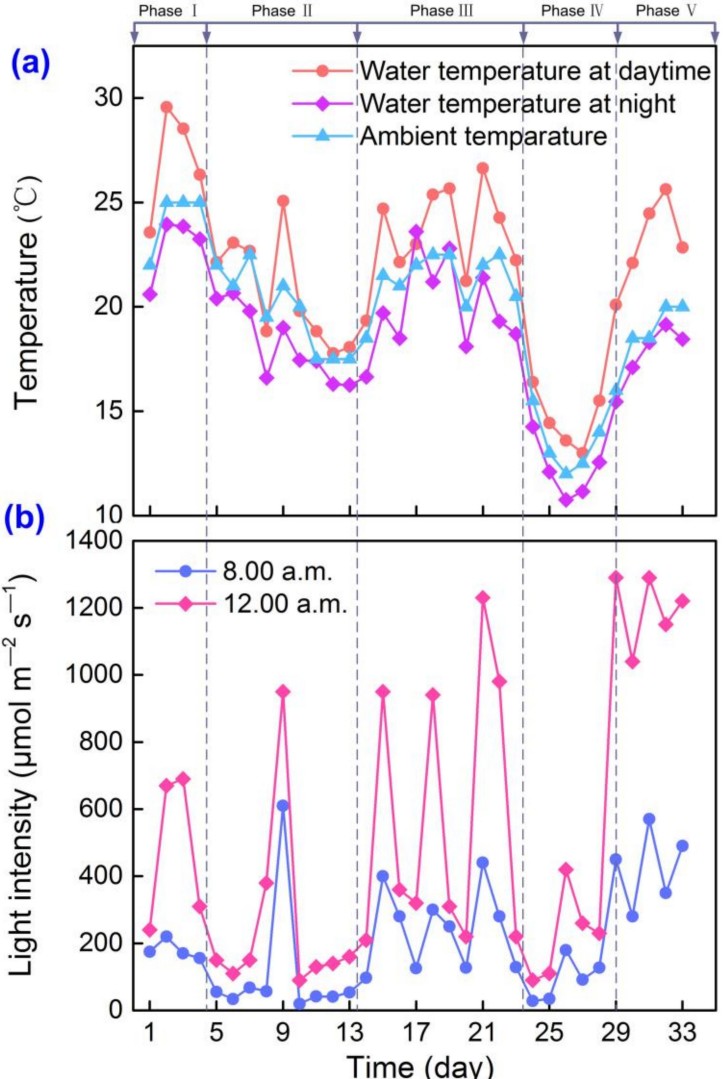

**Figure 2.** Experimental conditions in terms of average ambient temperature, water temperature (**a**) and light intensity (**b**).

The average removal rates of $PO_4^{3-}$-P at daytime and night were 45.1% $\pm$ 27.2% and 46.8% $\pm$ 22.2% with an insignificant difference ($p > 0.05$) (Figure 3c), showing no correlation with the average water temperature and light intensity (ANOVA, $p > 0.05$). In phase I, $PO_4^{3-}$-P removal at night was much better than that at daytime ($p < 0.05$), while insignificantly different in phase II ($p > 0.05$). Due to the unstable operation, $PO_4^{3-}$-P removal changed significantly ($p > 0.05$). In addition, little $PO_4^{3-}$-P removal was achieved during the days with high light intensity.

As can be seen from the above results, COD, $NH_4^+$-N and $PO_4^{3-}$-P could be effectively removed by MBGS at outdoor conditions. Due to lower temperature and absent light irradiation at night, removal rates of COD and $PO_4^{3-}$-P were lower at night than that at daytime, but the difference was insignificant. Different from COD and $PO_4^{3-}$-P removal, $NH_4^+$-N removal was significantly lower at night, indicating that $NH_4^+$-N was greatly

affected by temperature and illumination. As for the actual wastewater treatment process in the presence of MBGS, enhanced $NH_4^+$-N removal was necessary on cold or cloudy days.

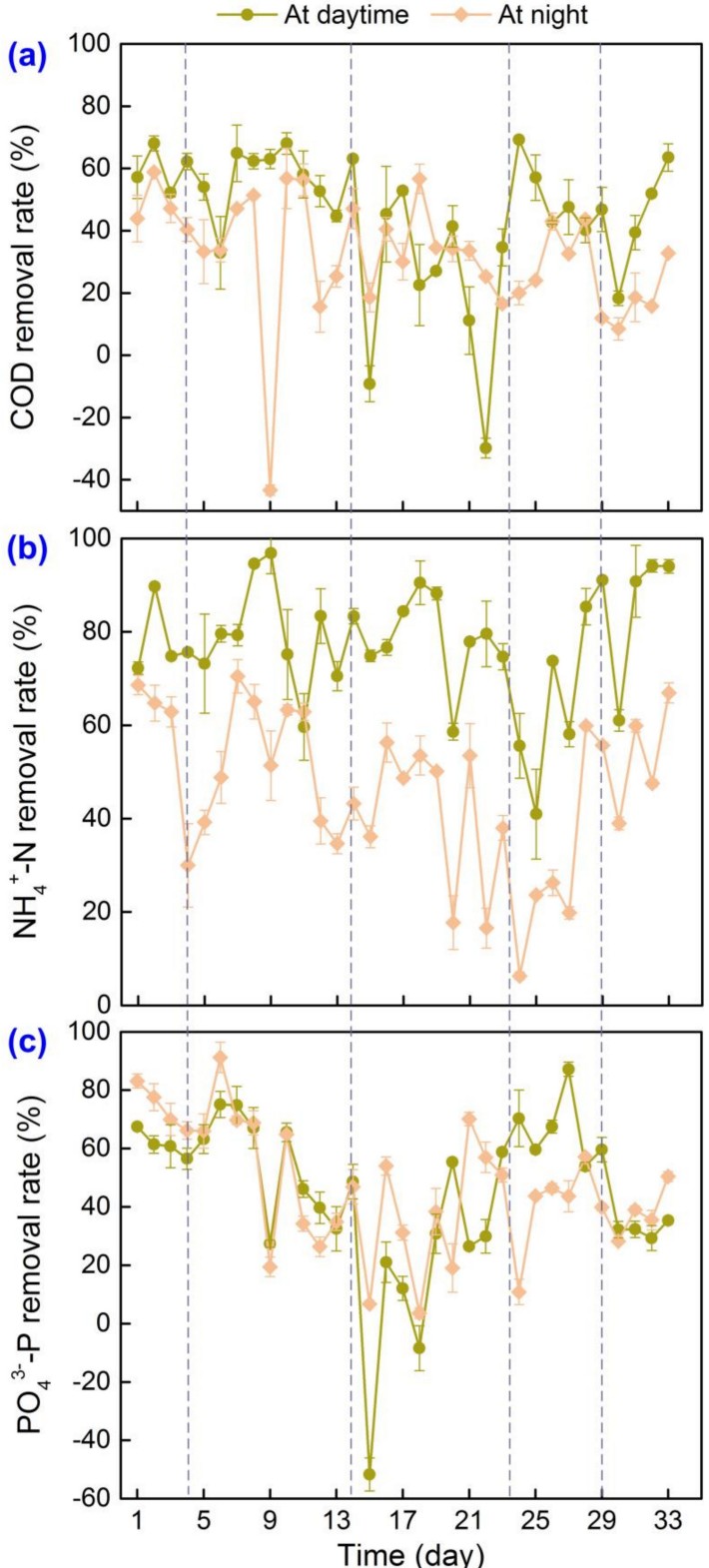

**Figure 3.** Removal rates of (**a**) COD, (**b**) $NH_4^+$-N and (**c**) $PO_4^{3-}$-P.

**Table 1.** The influent and effluent values in terms of Chemical oxygen demand (COD), $NH_4^+$-N and $PO_4^{3-}$-P.

| Time (day) | COD (mg/L) | | | | $NH_4^+$-N (mg/L) | | | | $PO_4^{3-}$-P (mg/L) | | | |
|---|---|---|---|---|---|---|---|---|---|---|---|---|
| | Daytime | | Night | | Daytime | | Night | | Daytime | | Night | |
| | Influent | Effluent | Influent | Effluent | Influent | Effluent | Influent | Effluent | Influent | Effluent | Influent | Effluent |
| 1 | 415 ± 14 | 178 ± 34 | 477 ± 12 | 268 ± 29 | 40.1 ± 0.5 | 11.1 ± 0.7 | 35.1 ± 0.1 | 11.0 ± 0.7 | 6.83 ± 0.17 | 2.22 ± 0.02 | 6.15 ± 0.12 | 1.04 ± 0.17 |
| 2 | 393 ± 6 | 126 ± 11 | 510 ± 70 | 210 ± 29 | 36.9 ± 0.1 | 3.8 ± 0.1 | 63.3 ± 6.2 | 22.3 ± 10.7 | 5.45 ± 0.34 | 2.11 ± 0.03 | 6.26 ± 0.27 | 1.41 ± 0.23 |
| 3 | 352 ± 40 | 168 ± 14 | 525 ± 67 | 276 ± 12 | 36.3 ± 1.9 | 9.2 ± 0.5 | 37.2 ± 3.1 | 13.8 ± 0.1 | 6.13 ± 0.05 | 2.41 ± 0.47 | 8.37 ± 0.63 | 2.52 ± 0.27 |
| 4 | 490 ± 57 | 184 ± 9 | 442 ± 9 | 263 ± 12 | 45.7 ± 2.5 | 11.1 ± 0.9 | 35.0 ± 1.7 | 24.5 ± 1.9 | 5.77 ± 0.64 | 2.51 ± 0.48 | 4.09 ± 0.19 | 1.38 ± 0.18 |
| 5 | 417 ± 63 | 190 ± 11 | 303 ± 3 | 202 ± 29 | 26.6 ± 10.6 | 7.1 ± 0.2 | 25.8 ± 1.5 | 15.7 ± 1.6 | 3.56 ± 0.72 | 1.31 ± 0.44 | 2.23 ± 0.08 | 0.76 ± 0.10 |
| 6 | 302 ± 37 | 205 ± 60 | 455 ± 26 | 300 ± 1 | 24.0 ± 0.8 | 4.9 ± 0.3 | 29.4 ± 2.1 | 15.0 ± 0.6 | 2.26 ± 0.86 | 0.56 ± 0.31 | 3.86 ± 0.29 | 0.34 ± 0.23 |
| 7 | 458 ± 29 | 162 ± 52 | 362 ± 6 | 191 ± 3 | 29.7 ± 2.3 | 6.1 ± 1.1 | 23.4 ± 0.2 | 6.9 ± 0.9 | 2.72 ± 0.02 | 0.69 ± 0.17 | 2.01 ± 0.05 | 0.61 ± 0.04 |
| 8 | 369 ± 74 | 140 ± 37 | 385 ± 84 | 187 ± 44 | 28.2 ± 1.9 | 1.5 ± 0.1 | 29.7 ± 0.8 | 10.4 ± 1.4 | 2.71 ± 0.63 | 0.90 ± 0.39 | 7.61 ± 0.12 | 2.37 ± 0.35 |
| 9 | 373 ± 1 | 138 ± 11 | 305 ± 12 | 436 ± 12 | 30.3 ± 0.3 | 0.9 ± 1.3 | 29.5 ± 1.2 | 14.3 ± 2.8 | 6.83 ± 0.50 | 4.96 ± 0.28 | 6.72 ± 0.03 | 5.41 ± 0.25 |
| 10 | 340 ± 34 | 109 ± 23 | 473 ± 31 | 189 ± 87 | 25.1 ± 0.6 | 6.2 ± 2.3 | 30.8 ± 2.0 | 11.3 ± 0.4 | 5.89 ± 0.23 | 2.03 ± 0.27 | 7.96 ± 0.18 | 2.80 ± 0.07 |
| 11 | 450 ± 6 | 188 ± 32 | 342 ± 47 | 150 ± 38 | 27.5 ± 0.4 | 11.1 ± 2.3 | 20.8 ± 0.1 | 7.7 ± 0.4 | 6.74 ± 0.03 | 3.63 ± 0.18 | 5.72 ± 0.36 | 3.76 ± 0.09 |
| 12 | 253 ± 9 | 120 ± 9 | 241 ± 15 | 204 ± 32 | 16.9 ± 0.9 | 2.8 ± 0.8 | 21.5 ± 0.9 | 13.0 ± 0.5 | 5.01 ± 0.23 | 3.02 ± 0.13 | 5.54 ± 0.29 | 4.08 ± 0.02 |
| 13 | 334 ± 37 | 184 ± 14 | 237 ± 20 | 177 ± 23 | 17.2 ± 0.4 | 5.1 ± 0.6 | 30.7 ± 1.7 | 18.7 ± 0.4 | 6.03 ± 0.53 | 4.07 ± 0.10 | 5.22 ± 0.27 | 3.40 ± 0.02 |
| 14 | 642 ± 117 | 237 ± 49 | 469 ± 35 | 247 ± 12 | 40.5 ± 1.4 | 6.8 ± 0.4 | 35.1 ± 1.5 | 19.9 ± 0.4 | 8.20 ± 0.64 | 4.22 ± 0.16 | 6.40 ± 0.28 | 3.40 ± 0.22 |
| 15 | 298 ± 32 | 324 ± 17 | 354 ± 6 | 288 ± 12 | 29.3 ± 0.2 | 7.4 ± 0.3 | 27.3 ± 5.6 | 17.5 ± 2.9 | 4.18 ± 0.13 | 6.33 ± 0.04 | 4.92 ± 0.02 | 4.58 ± 0.05 |
| 16 | 338 ± 20 | 186 ± 63 | 486 ± 41 | 288 ± 6 | 28.9 ± 1.1 | 6.7 ± 0.7 | 27.6 ± 1.1 | 12.1 ± 1.2 | 4.44 ± 0.19 | 3.51 ± 0.16 | 8.02 ± 0.23 | 3.69 ± 0.14 |
| 17 | 472 ± 37 | 223 ± 23 | 268 ± 64 | 185 ± 29 | 26.4 ± 0.2 | 4.1 ± 0.1 | 22.7 ± 3.3 | 11.7 ± 1.9 | 6.72 ± 0.03 | 5.91 ± 0.25 | 6.33 ± 0.58 | 4.36 ± 0.24 |
| 18 | 324 ± 29 | 249 ± 20 | 278 ± 44 | 121 ± 32 | 21.6 ± 0.8 | 2.0 ± 0.9 | 20.6 ± 0.9 | 9.6 ± 0.4 | 6.04 ± 0.27 | 6.55 ± 0.17 | 5.61 ± 0.05 | 5.41 ± 0.15 |
| 19 | 278 ± 37 | 203 ± 29 | 292 ± 29 | 191 ± 20 | 36.1 ± 0.8 | 4.2 ± 0.4 | 25.1 ± 2.2 | 12.5 ± 1.0 | 7.43 ± 0.37 | 5.14 ± 0.24 | 5.11 ± 0.45 | 3.15 ± 0.13 |
| 20 | 393 ± 103 | 227 ± 34 | 259 ± 6 | 171 ± 15 | 26.7 ± 0.7 | 11.0 ± 0.8 | 27.8 ± 1.4 | 15.1 ± 0.6 | 4.64 ± 0.07 | 2.08 ± 0.01 | 2.42 ± 0.35 | 1.96 ± 0.08 |
| 21 | 257 ± 32 | 227 ± 1 | 226 ± 6 | 150 ± 3 | 38.4 ± 0.3 | 8.5 ± 0.2 | 34.4 ± 1.3 | 16.0 ± 3.0 | 5.51 ± 0.02 | 4.05 ± 0.04 | 4.68 ± 0.08 | 1.40 ± 0.13 |
| 22 | 203 ± 29 | 263 ± 31 | 292 ± 41 | 218 ± 29 | 41.3 ± 12.6 | 8.4 ± 0.2 | 23.3 ± 0.4 | 19.4 ± 0.7 | 3.34 ± 0.19 | 2.34 ± 0.30 | 5.22 ± 0.97 | 2.25 ± 0.69 |
| 23 | 340 ± 22 | 221 ± 6 | 286 ± 14 | 239 ± 14 | 40.7 ± 1.3 | 10.3 ± 1.5 | 33.2 ± 0.7 | 20.6 ± 0.4 | 5.98 ± 0.75 | 2.47 ± 0.31 | 5.64 ± 0.19 | 2.77 ± 0.04 |
| 24 | 385 ± 3 | 197 ± 11 | 251 ± 64 | 199 ± 42 | 33.3 ± 3.2 | 14.8 ± 0.9 | 26.6 ± 0.4 | 24.9 ± 0.6 | 4.20 ± 0.19 | 1.25 ± 0.35 | 3.09 ± 0.25 | 2.75 ± 0.09 |
| 25 | 349 ± 53 | 148 ± 3 | 197 ± 6 | 150 ± 6 | 27.2 ± 1.2 | 16.1 ± 1.9 | 39.2 ± 0.4 | 29.9 ± 0.4 | 3.07 ± 0.23 | 1.24 ± 0.11 | 5.12 ± 0.21 | 2.89 ± 0.15 |
| 26 | 310 ± 3 | 178 ± 0 | 207 ± 20 | 118 ± 17 | 43.3 ± 1.3 | 11.4 ± 0.4 | 32.9 ± 1.5 | 24.2 ± 0.2 | 5.52 ± 0.23 | 1.80 ± 0.20 | 4.50 ± 0.29 | 2.41 ± 0.23 |
| 27 | 326 ± 25 | 172 ± 42 | 176 ± 8 | 118 ± 6 | 36.6 ± 1.9 | 15.3 ± 0.2 | 28.1 ± 1.2 | 22.5 ± 0.6 | 4.42 ± 0.09 | 0.57 ± 0.10 | 3.01 ± 0.01 | 1.70 ± 0.17 |
| 28 | 318 ± 31 | 190 ± 6 | 249 ± 1 | 140 ± 3 | 26.1 ± 0.1 | 3.8 ± 1.0 | 20.8 ± 0.8 | 8.3 ± 0.2 | 5.85 ± 0.08 | 2.70 ± 0.05 | 5.96 ± 0.01 | 2.55 ± 0.07 |
| 29 | 379 ± 56 | 199 ± 3 | 381 ± 20 | 336 ± 17 | 23.1 ± 0.3 | 2.1 ± 0.1 | 21.7 ± 0.1 | 9.6 ± 0.2 | 5.46 ± 0.19 | 2.21 ± 0.30 | 6.01 ± 0.12 | 3.61 ± 0.15 |
| 30 | 152 ± 20 | 124 ± 20 | 397 ± 3 | 363 ± 17 | 2.5 ± 0.1 | 1.0 ± 0.1 | 20.1 ± 0.5 | 12.3 ± 0.6 | 6.53 ± 0.50 | 4.42 ± 0.52 | 6.46 ± 0.18 | 4.64 ± 0.14 |
| 31 | 490 ± 17 | 296 ± 17 | 395 ± 89 | 318 ± 42 | 18.2 ± 6.8 | 1.7 ± 0.7 | 22.2 ± 0.2 | 8.9 ± 0.4 | 7.08 ± 0.16 | 4.79 ± 0.31 | 6.80 ± 0.04 | 4.14 ± 0.06 |
| 32 | 217 ± 11 | 52 ± 1 | 499 ± 20 | 421 ± 14 | 21.4 ± 0.1 | 1.3 ± 0.3 | 24.1 ± 0.3 | 12.6 ± 0.3 | 5.69 ± 0.01 | 4.02 ± 0.25 | 6.88 ± 0.03 | 4.44 ± 0.21 |
| 33 | 154 ± 17 | 64 ± 4 | 458 ± 56 | 308 ± 39 | 28.1 ± 0.3 | 1.7 ± 0.4 | 23.7 ± 0.6 | 7.8 ± 0.7 | 7.41 ± 0.17 | 4.79 ± 0.14 | 6.21 ± 0.01 | 3.08 ± 0.11 |

### 3.2. Basic Parameters of MBGS

The basic parameters of MBGS at the beginning and end of the experiment are presented in Table 2. After the operation, the granular size was reduced by 84.6 μm. This indicated that the stability of MBGS was likely destroyed due to blocked growth of microalgae and bacteria under changed outdoor ambient temperature and light intensity, leading to decreased removal rates of pollutants (Figure 3). On the other hand, $SVI_5$ increased by 9.5 mL/g. As bacterial sludge had a much better settling capacity than microalgae [24,25], microalgae may be reduced in MBGS in this study, evidenced by reduced chlorophyll a (i.e., 19.3 mg/g) and chlorophyll b (i.e., 10.3 mg/g). In fact, chlorophyll a was commonly used as an algae biomass indicator [26,27]. As such, microalgae in MBGS were reduced during the experiment. It should be noted that the ratio of chlorophyll a to chlorophyll b increased, possibly due to the elevation of cyanobacteria.

**Table 2.** Basic parameters of MBGS before and after the experiment.

| Time (d) | $SVI_5$ (mL/g) | Chlorophyll a (mg/g) | Chlorophyll b (mg/g) | Chlorophyll a/b | Granular Size (μm) |
|---|---|---|---|---|---|
| 1 | 73.6 ± 10.1 | 20.5 ± 0.7 | 10.7 ± 0.6 | 1.89 ± 0.04 | 710.2 ± 18.6 |
| 33 | 83.1 ± 0.1 | 1.2 ± 0.2 | 0.4 ± 0.1 | 2.99 ± 0.87 | 625.6 ± 28.4 |

### 3.3. The Community Structure of MBGS

The distribution of prokaryotes at class level and eukaryotes at genus level in MBGS were presented in Figure 4a,b. As seen, *Gammaproteobacteria* attributed to *Aquimonas*, *Acinetobacter*, *Pseudomonas*, *Thauera*, *Clostridia* attributed to *Proteiniclasticum*, *Acetoanaerobium*, *Alphaproteobacteria* attributed to *Rhizobium*, *Phreatobacter*, *Psychroglaciecola* and *Rhodobacter* were predominant prokaryotic microbial community, most of which were heterogenetic bacteria that oxidized organic pollutants and produced $CO_2$ for microalgae [28]. In addition, nitrifying bacteria were not detected in MBGS, and nitrates and nitrites were not found in water, indicating that $NH_4^+$-N removal was mainly due to microbial assimilation [12]. As such, nutrient removal at daytime was higher than that at night due to $O_2$ production from microalgal photosynthesis. On the other hand, common phosphate-accumulating organisms such as *Actinobacteria* (e.g., *Microlunatus* and *Tetrasphaera*), *Gammaproteobacteria* (e.g., *Halomonas* and *Thiothrix*), *Betaproterbacteria* (e.g., *Accumulibacter*, *Dechloromonas* and *Comamonadaceae*) and *Pantanalinema* were not the dominant bacteria in MBGS, indicating that $PO_4^{3-}$-P removal was mainly attributed to microbial assimilation [29]. Thus, poor $PO_4^{3-}$-P removal was obtained in this study (Figure 3c).

The relative abundance of *Aquimonas* on day 33 was 13.6% lower than that on day 17 (Figure 4b). The plausible reason was that *Aquimonas* was thermophilic and temperature in phase IV was the lowest. In addition, *Aquimonas* was likely to compete with denitrifying bacteria (i.e., *Pseudomonas*, *Thauera*, *Rhizobium*, *Phreatobacter*, *Psychroglaciecola*, *Rhodobacter*) [30–33]. The total abundance of these denitrifying bacteria increased from 13.0% to 18.0% after the whole experiment. The relative abundance of filamentous microorganisms (i.e., *Clostridia* and *Anaerolineae*) decreased from 35.7% on day 1 to 31.6% on day 17, which was helpful to improve the stability of granular structures [34,35], leading to smaller particle size (Table 1). On the other hand, *Chlorella* was the dominant genus of eukaryotes with an abundance of above 99% (Figure 4c), indicating that environmental changes insignificantly affected the dominance of *Chlorella*.

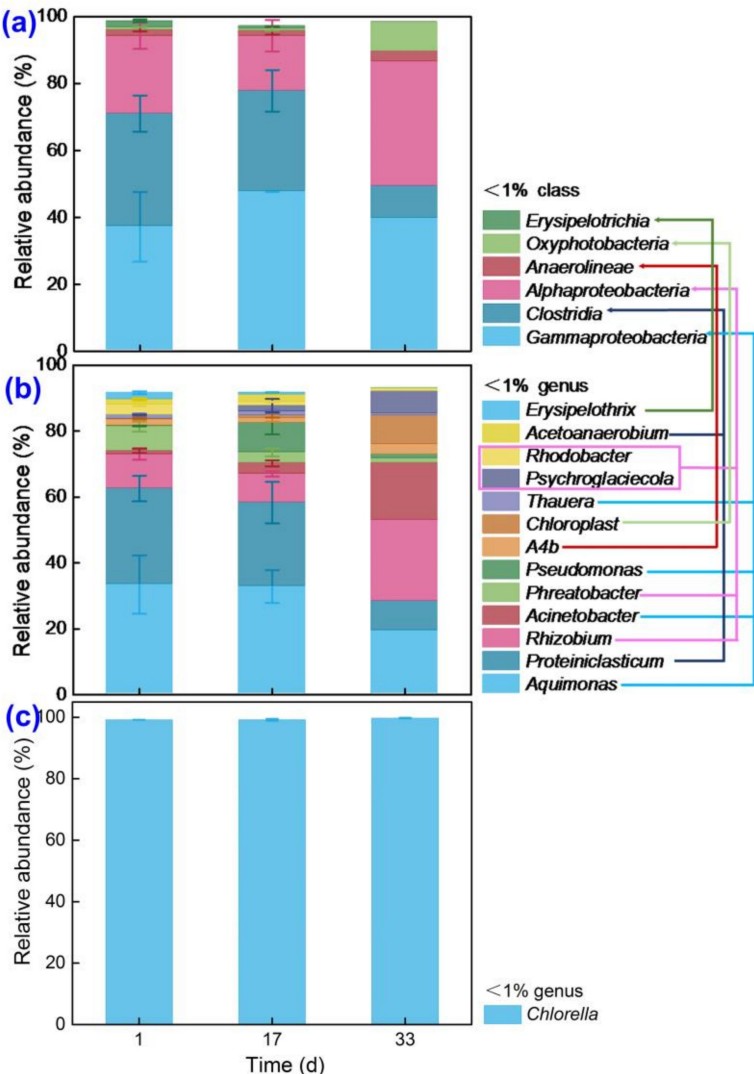

**Figure 4.** Distributions of prokaryotic diversity at class level (**a**) and genus level (**b**); eukaryotic diversity at genus level in microbial communities (**c**) on days 1, 17 and 33. Minority fractions of bacterial and algal communities indicated the sum of different classes and genus accounted for less than 1% of its total sequences in each sample.

## 4. Conclusions and Perspectives

In this study, COD, $NH_4^+$-N and $PO_4^{3-}$-P removal was investigated for the first time by microalgal-bacterial granular sludge (MBGS) process under actual 33 day-outdoor conditions experienced two cooling and lasted continuous rainy days. It was found that COD and $NH_4^+$-N removal was better at daytime than at night, while $PO_4^{3-}$-P removal was insignificantly different at daytime and night. In addition, COD and $PO_4^{3-}$-P fluctuated greatly with the operation of the MBGS system while they were of insignificant difference with water temperature and light intensity. However, $NH_4^+$-N removal was significantly related to water temperature while insignificantly correlated with light intensity. On the other hand, the prokaryotic community structure of MBGS changed significantly while *Chlorella* always maintained an absolutely dominant position in eukaryotes.

In view of the adaptability potential of the MBGS process to natural day-night alternation in wastewater treatment, it is expected to be the next generation of wastewater treatment technology. At present, the research of the MBGS process has made great progress, but there are still problems that need to be solved in the practical application. Future research should focus on the following aspects: (1) exploring a continuous-flow photo-bioreactor with good hydraulic conditions for real domestic wastewater treatment;

(2) the reuse of the produced biomass of MBGS; (3) the potential of MBGS for the removal of emerging contaminants. These would accelerate the application of the MBGS process in the application of wastewater reclamation.

**Author Contributions:** Conceptualization, B.J.; methodology, B.J. and S.W.; validation, Y.M.; investigation, L.Z.; data curation, L.Z. and H.H.; writing—original draft preparation, S.W.; writing—review and editing, B.J. and Y.M. All authors have read and agreed to the published version of the manuscript.

**Funding:** This research received no external funding.

**Institutional Review Board Statement:** Not applicable.

**Informed Consent Statement:** Informed consent was obtained from all subjects involved in the study.

**Data Availability Statement:** Not applicable.

**Conflicts of Interest:** The authors declare no conflict of interest.

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
