# Peer review of "Microalgal-Bacterial Granular Sludge Process in Non-Aerated Municipal Wastewater Treatment under Natural Day-Night Conditions: Performance and Microbial Community"

_water, doi:10.3390/w13111479_

Round 1

Reviewer 1 Report

The work is very interesting, it is well planned; the methodology is well described and the references adequate and current.

However, it lacks a more in-depth discussion on the effectiveness of the MBGS system and proposals for improvements to the system for its future full-scale application.
The conclusions as they are exposed seem more like results. They could be written in a single paragraph. Future research proposals could also be included.

I attach the revised manuscript with minors comments

Author Response

Thank you very much for your encouraging comments. Some in-depth discussions were added. In addition, the conclusions were re-written in single paragraphs including future research proposals.

Reviewer 2 Report

The article is interesting and provides important information for the scientific community and plant managers. The graphs and results are presented well. However the authors should improve the description of the microalgal-bacterial granular sludge (MBGS) process both in introducion and material and methods. As it is presented now it is not clear and the process is not easy to understand. I only suggest this review.

Author Response

Thank you for your kind suggestions and totally agreed with you. The descriptions of MBGS process was added in the introduction and Section 2.1.

Reviewer 3 Report

The authors are not looking to implement this technology in colder places where temperatures fall below freezing temperatures?

How was 50ml volume of glass anaerobic bottles chosen and was there any special reason for that?

The authors mention PO43--P at daytime and night was 45.1% ± 143 27.2% and 46.8% ± 22.1%. Would it be possible to provide initial and final values for those numbers as well as other parameters being measured?

Did the authors collect control sets of data which can serve as basis for the comparison of results?

Are the experiments performed in Batch mode? Is it necessary to conduct the continues mode of experiments which would more of close analogy to real world cases?

Author Response

  1. The authors are not looking to implement this technology in colder places where temperatures fall below freezing temperatures?

The microalgal-bacterial granular sludge (MBGS) process as a biological treatment technology was seriously restricted by below freezing temperatures. Therefore, this technology had not successfully implemented in colder places. However, our previous study indicated that MBGS process could perform satisfactorily in the temperature ranging from 15 to 30 â—¦C (Bin Ji, Lin Zhu, Shulian Wang, Yu Liu. Temperature-effect on the performance of non-aerated microalgal-bacterial granular sludge process in municipal wastewater treatment. Journal of Environmental Management, 2021, 282, 111955).

  1. How was 50ml volume of glass anaerobic bottles chosen and was there any special reason for that?

Our previous researches were carried out using 50-60 ml of glass bottles (25-30 ml of working volume) (Bin Ji, Meng Zhang, Jun Gu, Yingqun Ma, Yu Liu. A self-sustaining synergetic microalgal-bacterial granular sludge process towards energy-efficient and environmentally sustainable municipal wastewater treatment. Water Research, 2020, 179, 115884; Shulian Wang, Bin Ji, Baihui Cui, Yingqun Ma, Dabin Guo, Yu Liu. Cadmium-effect on performance and symbiotic relationship of microalgal-bacterial granules. Journal of Cleaner Production, 2021, 282, 125383; Shulian Wang, Bin Ji, Meng Zhang, Yingqun Ma, Jun Gu, Yu Liu. Defensive responses of microalgal-bacterial granules to tetracycline in municipal wastewater treatment. Bioresource Technology, 2020, 312, 123605). Therefore, 50 ml volume of glass anaerobic bottles was chosen in this study.

  1. The authors mention PO43--P at daytime and night was 45.1% ± 27.2% and 46.8% ± 22.1%. Would it be possible to provide initial and final values for those numbers as well as other parameters being measured?

The influent and effluent values in terms of COD, NH4+-N and PO43--P were presented in the following table.

Time (day)

COD (mg/L)

NH4+-N (mg/L)

PO43--P (mg/L)

Daytime

Night

Daytime

Night

Daytime

Night

Influent

Effluent

Influent

Effluent

Influent

Effluent

Influent

Effluent

Influent

Effluent

Influent

Effluent

1

415±14

178±34

477±12

268±29

40.1±0.5

11.1±0.7

35.1±0.1

11.0±0.7

6.83±0.17

2.22±0.02

6.15±0.12

1.04±0.17

2

393±6

126±11

510±70

210±29

36.9±0.1

3.8±0.1

63.3±6.2

22.3±10.7

5.45±0.34

2.11±0.03

6.26±0.27

1.41±0.23

3

352±40

168±14

525±67

276±12

36.3±1.9

9.2± 0.5

37.2±3.1

13.8±0.1

6.13±0.05

2.41±0.47

8.37±0.63

2.52±0.27

4

490±57

184±9

442±9

263±12

45.7±2.5

11.1±0.9

35.0±1.7

24.5±1.9

5.77±0.64

2.51±0.48

4.09±0.19

1.38±0.18

5

417±63

190±11

303±3

202±29

26.6±10.6

7.1±0.2

25.8±1.5

15.7±1.6

3.56±0.72

1.31±0.44

2.23±0.08

0.76±0.10

6

302±37

205±60

455±26

300±1

24.0±0.8

4.9±0.3

29.4±2.1

15.0±0.6

2.26±0.86

0.56±0.31

3.86±0.29

0.34±0.23

7

458±29

162±52

362±6

191±3

29.7±2.3

6.1±1.1

23.4±0.2

6.9±0.9

2.72±0.02

0.69±0.17

2.01±0.05

0.61±0.04

8

369±74

140±37

385±84

187±44

28.2±1.9

1.5±0.1

29.7±0.8

10.4±1.4

2.71±0.63

0.90±0.39

7.61±0.12

2.37±0.35

9

373±1

138±11

305±12

436±12

30.3±0.3

0.9±1.3

29.5±1.2

14.3±2.8

6.83±0.50

4.96±0.28

6.72±0.03

5.41±0.25

10

340±34

109±23

473±31

189±87

25.1±0.6

6.2±2.3

30.8±2.0

11.3±0.4

5.89±0.23

2.03±0.27

7.96±0.18

2.80±0.07

11

450±6

188±32

342±47

150±38

27.5±0.4

11.1±2.3

20.8±0.1

7.7±0.4

6.74±0.03

3.63±0.18

5.72±0.36

3.76±0.09

12

253±9

120±9

241±15

204±32

16.9±0.9

2.8±0.8

21.5±0.9

13.0±0.5

5.01±0.23

3.02±0.13

5.54±0.29

4.08±0.02

13

334±37

184±14

237±20

177±23

17.2±0.4

5.1±0.6

30.7±1.7

18.7±0.4

6.03±0.53

4.07±0.10

5.22±0.27

3.40±0.02

14

642±117

237±49

469±35

247±12

40.5±1.4

6.8±0.4

35.1±1.5

19.9±0.4

8.20±0.64

4.22±0.16

6.40±0.28

3.40±0.22

15

298±32

324±17

354±6

288±12

29.3±0.2

7.4±0.3

27.3±5.6

17.5±2.9

4.18±0.13

6.33±0.04

4.92±0.02

4.58±0.05

16

338±20

186±63

486±41

288±6

28.9±1.1

6.7±0.7

27.6±1.1

12.1±1.2

4.44±0.19

3.51±0.16

8.02±0.23

3.69±0.14

17

472±37

223±23

268±64

185±29

26.4±0.2

4.1±0.1

22.7±3.3

11.7±1.9

6.72±0.03

5.91±0.25

6.33±0.58

4.36±0.24

18

324±29

249±20

278±44

121±32

21.6±0.8

2.0±0.9

20.6±0.9

9.6±0.4

6.04±0.27

6.55±0.17

5.61±0.05

5.41±0.15

19

278±37

203±29

292±29

191±20

36.1±0.8

4.2±0.4

25.1±2.2

12.5±1.0

7.43±0.37

5.14±0.24

5.11±0.45

3.15±0.13

20

393±103

227±34

259±6

171±15

26.7±0.7

11.0±0.8

27.8±1.4

15.1±0.6

4.64±0.07

2.08±0.01

2.42±0.35

1.96±0.08

21

257±32

227±1

226±6

150±3

38.4±0.3

8.5±0.2

34.4±1.3

16.0±3.0

5.51±0.02

4.05±0.04

4.68±0.08

1.40±0.13

22

203±29

263±31

292±41

218±29

41.3±12.6

8.4±0.2

23.3±0.4

19.4±0.7

3.34±0.19

2.34±0.30

5.22±0.97

2.25±0.69

23

340±22

221±6

286±14

239±14

40.7±1.3

10.3±1.5

33.2±0.7

20.6±0.4

5.98±0.75

2.47±0.31

5.64±0.19

2.77±0.04

24

385±3

197±11

251±64

199±42

33.3±3.2

14.8±0.9

26.6±0.4

24.9±0.6

4.20±0.19

1.25±0.35

3.09±0.25

2.75±0.09

25

349±53

148±3

197±6

150±6

27.2±1.2

16.1±1.9

39.2±0.4

29.9±0.4

3.07±0.23

1.24±0.11

5.12±0.21

2.89±0.15

26

310±3

178±0

207±20

118±17

43.3±1.3

11.4±0.4

32.9±1.5

24.2±0.2

5.52±0.23

1.80±0.20

4.50±0.29

2.41±0.23

27

326±25

172±42

176±8

118±6

36.6±1.9

15.3±0.2

28.1±1.2

22.5±0.6

4.42±0.09

0.57±0.10

3.01±0.01

1.70±0.17

28

318±31

190±6

249±1

140±3

26.1±0.1

3.8±1.0

20.8±0.8

8.3±0.2

5.85±0.08

2.70±0.05

5.96±0.01

2.55±0.07

29

379±56

199±3

381±20

336±17

23.1±0.3

2.1±0.1

21.7±0.1

9.6±0.2

5.46±0.19

2.21±0.30

6.01±0.12

3.61±0.15

30

152±20

124±20

397±3

363±17

2.5±0.1

1.0±0.1

20.1±0.5

12.3±0.6

6.53±0.50

4.42±0.52

6.46±0.18

4.64±0.14

31

490±17

296±17

395±89

318±42

18.2±6.8

1.7±0.7

22.2±0.2

8.9±0.4

7.08±0.16

4.79±0.31

6.80±0.04

4.14±0.06

32

217±11

52±1

499±20

421±14

21.4±0.1

1.3±0.3

24.1±0.3

12.6±0.3

5.69±0.01

4.02±0.25

6.88±0.03

4.44±0.21

33

154±17

64±4

458±56

308±39

28.1±0.3

1.7±0.4

23.7±0.6

7.8±0.7

7.41±0.17

4.79±0.14

6.21±0.01

3.08±0.11

  1. Did the authors collect control sets of data which can serve as basis for the comparison of results?

The control experiment in the absence of MBGS under identical conditions had been carried out and no nutrients removal was found.

  1. Are the experiments performed in Batch mode? Is it necessary to conduct the continues mode of experiments which would more of close analogy to real world cases?

It is true the experiment was performed in batch mode. It is necessary to conduct continues mode and we are doing it now, which also shows good performances.